# Membranes, energetics, and evolution across the prokaryote-eukaryote divide

Michael Lynch*, Georgi K Marinov

Department of Biology, Indiana University, Bloomington, United States

**Abstract** The evolution of the eukaryotic cell marked a profound moment in Earth's history, with most of the visible biota coming to rely on intracellular membrane-bound organelles. It has been suggested that this evolutionary transition was critically dependent on the movement of ATP synthesis from the cell surface to mitochondrial membranes and the resultant boost to the energetic capacity of eukaryotic cells. However, contrary to this hypothesis, numerous lines of evidence suggest that eukaryotes are no more bioenergetically efficient than prokaryotes. Thus, although the origin of the mitochondrion was a key event in evolutionary history, there is no reason to think membrane bioenergetics played a direct, causal role in the transition from prokaryotes to eukaryotes and the subsequent explosive diversification of cellular and organismal complexity.

DOI: https://doi.org/10.7554/eLife.20437.001

## Introduction

The hallmark feature distinguishing eukaryotes from prokaryotes (bacteria and archaea) is the universal presence in the former of discrete cellular organelles enveloped within lipid bilayers (e.g. the nucleus, mitochondria, endoplasmic reticulum, golgi, vacuoles, vesicles, etc.). Under a eukaryocentric view of life, these types of cellular features promoted the secondary origin of genomic modifications that ultimately led to the adaptive emergence of fundamentally superior life forms (*Martin and Koonin, 2006*; *Lane and Martin, 2010*). Most notably, it has been proposed that the establishment of the mitochondrion provided an energetic boost that fueled an evolutionary revolution responsible for all things eukaryotic, including novel protein folds, membrane-bound organelles, sexual reproduction, multicellularity, and complex behavior (*Lane, 2002*, *2015*).

However, despite having more than two billion years to impose their presumed superiority, eukaryotes have not driven prokaryotes extinct. Prokaryotes dominate eukaryotes both on a numerical and biomass basis (*Whitman et al., 1998*; *Lynch, 2007*), and harbor most of the biosphere's metabolic diversity. Although there is no logical basis for proclaiming the evolutionary inferiority of prokaryotes, one central issue can be addressed objectively – the degree to which the establishment of eukaryotic-specific morphology altered energetic efficiency at the cellular level.

Drawing on observations from biochemistry, physiology, and cell biology, we present a quantitative summary of the relative bioenergetic costs and benefits of the modified architecture of the eukaryotic cell. The data indicate that once cell-size scaling is taken into account, the bioenergetic features of eukaryotic cells are consistent with those in bacteria. This implies that the mitochondrion-host cell consortium that became the primordial eukaryote did not precipitate a bioenergetics revolution.

## Results

### The energetic costs of building and maintaining a cell

The starting point is a recap of recent findings on the scaling properties of the lifetime energetic expenditures of single cells. All energy utilized by cells can be partitioned into two basic categories: that employed in cell maintenance and that directly invested in building the physical infrastructure

*For correspondence:
milynch@indiana.edu

Competing interests: The authors declare that no competing interests exist.

**eLife digest** Over time, life on Earth has evolved into three large groups: archaea, bacteria, and eukaryotes. The most familiar forms of life – such as fungi, plants and animals – all belong to the eukaryotes. Bacteria and archaea are simpler, single-celled organisms and are collectively referred to as prokaryotes.

The hallmark feature that distinguishes eukaryotes from prokaryotes is that eukaryotic cells contain compartments called organelles that are surrounded by membranes. Each organelle supports different activities in the cell. Mitochondria, for example, are organelles that provide eukaryotes with most of their energy by producing energy-rich molecules called ATP. Prokaryotes lack mitochondria and instead produce their ATP on their cell surface membrane.

Some researchers have suggested that mitochondria might actually be one of the reasons that eukaryotic cells are typically larger than prokaryotes and more varied in their shape and structure. The thinking is that producing ATP on dedicated membranes inside the cell, rather than on the cell surface, boosted the amount of energy available to eukaryotic cells and allowed them to diversify more. However, other researchers are not convinced by this view. Moreover, some recent evidence suggested that eukaryotes are no more efficient in producing energy than prokaryotes.

Lynch and Marinov have now used computational and comparative analysis to compare the energy efficiency of different organisms including prokaryotes and eukaryotes grown under defined conditions. To do the comparison, the results were scaled based on cell volume and the total surface area deployed in energy production.

From their findings, Lynch and Marinov concluded that mitochondria did not enhance how much energy eukaryotes could produce per unit of cell volume in any substantial way. Although the origin of mitochondria was certainly a key event in evolutionary history, it is unlikely to have been responsible for the diversity and complexity of today's life forms.
DOI: https://doi.org/10.7554/eLife.20437.002

that comprises a daughter cell. Maintenance costs involve a diversity of cellular functions, ranging from turnover of biomolecules, intracellular transport, control of osmotic balance and membrane potential, nutrient uptake, information processing, and motility. Cell growth represents a one-time investment in the production of the minimum set of parts required for a progeny cell, whereas cell maintenance costs scale with cell-division time. The common usage of metabolic rate as a measure of power production is uninformative from an evolutionary perspective, as it fails to distinguish the investment in cellular reproduction from that associated with non-growth-related processes.

To make progress in this area, a common currency of energy is required. The number of ATP→ADP turnovers meets this need, as such transformations are universally deployed in most cellular processes of all organisms, and where other cofactors are involved, these can usually be converted into ATP equivalents (*Atkinson, 1970*). When cells are grown on a defined medium for which the conversion rate from carbon source to ATP is known (from principles of biochemistry), the two categories of energy allocation can be quantified from the regression of rates of resource consumption per cell on rates of cell division (*Bauchop and Elsden, 1960*; *Pirt, 1982*; *Tempest and Neijssel, 1984*).

A summary of results derived from this method reveals two universal scaling relationships that transcend phylogenetic boundaries (*Lynch and Marinov, 2015*). First, basal maintenance costs (extrapolated to zero-growth rate, in units of $10^9$ molecules of ATP/cell/hour, and normalized to a constant temperature of 20°C for all species) scale with cell volume as a power-law relationship

$$C_M = 0.39V^{0.88}, \tag{1a}$$

where cell volume $V$ is in units of $\mu m^3$. Second, the growth requirements per cell (in units of $10^9$ molecules of ATP/cell) scale as

$$C_G = 27V^{0.97}. \tag{1b}$$

The total cost of building a cell is

$$C_T = C_G + t C_M, \tag{1c}$$

where $t$ is the cell-division time in hours.

Derived from cells ranging over four orders of magnitude in volume, neither of the preceding scaling relationships is significantly different from expectations under isometry (with exponent 1.0), as the standard errors of the exponents in *Equations (1a,b)* are 0.07 and 0.04, respectively. Moreover, as there is no discontinuity in scaling between prokaryotes and eukaryotes, these results suggest that a shift of bioenergetics from the cell membrane in prokaryotes to the mitochondria of eukaryotes conferred no directly favorable energetic effects. In fact, the effect appears to be negative.

Taking into account the interspecific relationships between cell-division time and cell volume (*Lynch and Marinov, 2015*) and using *Equation (1b)*, one can compute the scaling of the rate of incorporation of energy into biomass, $C_G/t$. For bacteria, cell-division times decline with increasing cell volume as $\sim V^{-0.17}$, albeit weakly (the SE of the exponent being 0.11), implying that the rate of biomass accumulation scales as $\sim V^{0.97+0.17} = V^{1.14}$ on a per-cell basis and as $\sim V^{1.14-1.00} = V^{0.14}$ on a cell volumetric basis (with the SEs of both exponents being 0.12). In contrast, in most eukaryotic groups, cell-division times increase with cell volume, on average scaling as $\sim V^{0.13}$, implying a scaling of $\sim V^{0.84}$ for the rate of biomass accumulation per cell and $\sim V^{-0.16}$ on a volumetric basis (with SEs equal to 0.06 for the exponents). Thus, in terms of biomass production, the bioenergetic efficiency of eukaryotic cells declines with cell volume, whereas that of bacterial cells does not. The pattern observed in bacteria is inconsistent with the view that surface area limits the rate of energy production, as this leads to an expected scaling of $\sim V^{2/3}$ on a per-cell basis.

## Energy production and the mitochondrion

The argument that mitochondria endow eukaryotic cells with exceptionally high energy provisioning derives from the idea that large internal populations of small mitochondria with high surface area-to-volume ratios provide a dramatic increase in bioenergetic-membrane capacity (*Lane and Martin, 2010*). In prokaryotes, the $F_0F_1$ ATP synthase (the molecular machine that transforms ADP to ATP in the process of chemiosmosis) and the electron transport chain (ETC) components (which create the chemiosmotic proton gradient) are restricted to the cell membrane, but in eukaryotes, they are confined to inner mitochondrial membranes. A key question is whether the bioenergetic capacity of cells is, in fact, limited by membrane surface area.

Although the situation at the time of first colonization of the mitochondrion is unknown, the iconic view of mitochondria being tiny, bean-shaped cellular inclusions is not entirely generalizable. For example, many unicellular eukaryotes harbor just a single mitochondrion or one that developmentally moves among alternative reticulate states (e.g. *Rosen et al., 1974*; *Osafune et al., 1975*; *Biswas et al., 2003*; *Yamaguchi et al., 2011*). Such geometries necessarily have lower total surface areas than a collection of spheroids with similar total volumes. For the range of species that have been examined, many of which do have small individualized mitochondria, the total outer surface area of mitochondria per cell is generally on the order of the total area of the plasma membrane, with no observed ratio exceeding 5:1, and many being considerably smaller than 1:1 (*Figure 1a*). It may be argued that the outer surface area of the mitochondrion is of less relevance than that of the inner membrane (where the ATP synthase complex sits), but the ratios of inner (including the internal cristae) to outer membrane areas for mitochondria in mammals, the green alga *Ochromonas*, the plant *Rhus toxicodendron*, and the ciliate *Tetrahymena* are 5.0 (SE = 1.1), 2.4, 2.5, and 5.2, respectively (Supplementary material). Thus, the data are inconsistent with the idea that the mitochondrion engendered a massive expansion in the surface area of bioenergetic membranes in eukaryotes.

Three additional observations raise questions as to whether membrane surface area is a limiting factor in ATP synthesis. First, the localization of mitochondrial ATP synthase complexes is restricted to two rows on the narrow edges of the inner cristae (*Kühlbrandt, 2015*). Because this confined region comprises <<10% of the total internal membrane area, the surface area of mitochondrial membranes allocated to ATP synthase appears to be less than the surface area of the cell itself. Second, only a fraction of bacterial membranes appears to be allocated to bioenergetic functions (*Magalon and Alberge, 2016*), again shedding doubt on whether membrane area is a limiting factor for energy production. Third, in every bacterial species for which data are available, growth in cell volume is close to exponential, that is, the growth rate of a cell increases as its cell volume increases

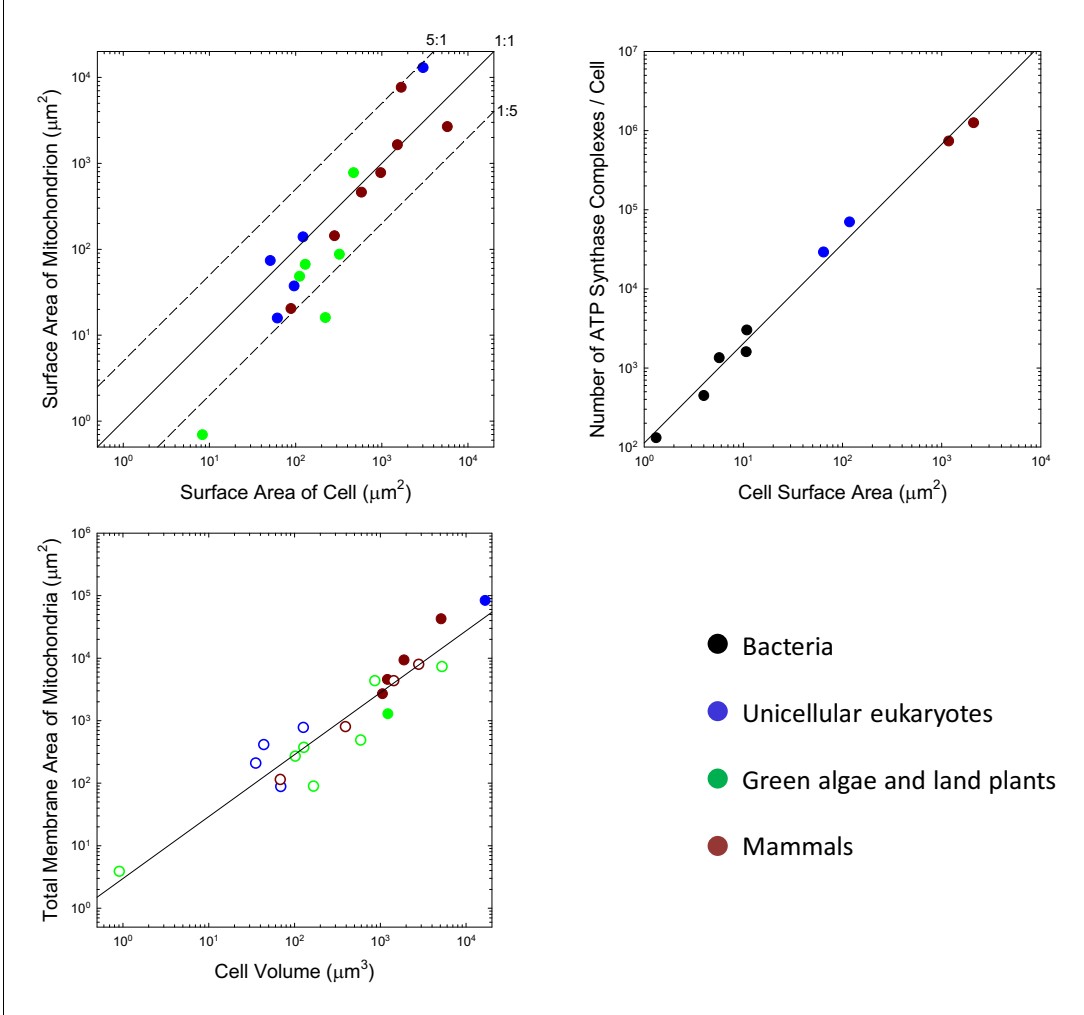

**Figure 1.** Scaling features of membrane properties with cell size. (a) Relationship between the total outer surface area of mitochondria and that of the plasma membrane for all species with available data. Diagonal lines denote three idealized ratios of the two. (b) The number of ATP synthase complexes per cell scales with cell surface area (S, in $\mu m^2$) as $113S^{1.26}$ ($r^2 = 0.99$). (c) Relationship between the total (inner + outer) surface area of mitochondria and cell volume for all species with available data. Open points are extrapolations for species with only outer membrane measures, derived by assuming an inner:outer ratio of 4.6, the average of observations in other species. References to individual data points are provided in *Appendix 1–tables 1* and *2*.
DOI: https://doi.org/10.7554/eLife.20437.003

despite the reduction in the surface area:volume ratio (*Voorn and Koppes, 1998*; *Godin et al., 2010*; *Santi et al., 2013*; *Iyer-Biswas et al., 2014*; *Osella et al., 2014*; *Campos et al., 2014*).

Further insight into this issue can be achieved by considering the average packing density of ATP synthase for the few species with proteomic data sufficient for single-cell counts of individual proteins. By accounting for the stoichiometry of the various subunits in the complex, it is possible to obtain several independent estimates of the total number of complexes per cell under the assumption that all the proteins are assembled (Supplementary material). For example, the estimated number of complexes in *E. coli* is 3018, and the surface area of the cell is ~15.8 $\mu m^2$. Based on the largest diameter of the molecule (the $F_1$ subcomplex), a single ATP synthase in this species occupies ~64 $nm^2$ (*Lücken et al., 1990*) of surface area, so the total set of complexes occupies ~1.8% of the cell membrane. Four other diverse bacterial species for which these analyses can be performed yield occupancies ranging from 0.6% to 1.5%, for an overall average of 1.1% for bacteria. This will be an

overestimate if only a fraction of proteins are properly assembled and embedded in the cell membrane.

This kind of analysis can be extended to eukaryotes, noting that eukaryotic ATP synthases are slightly larger, with maximum surface area of ~110 nm$^2$ (*Abrahams et al., 1994*; *Stock et al., 1999*). Although ATP synthase resides in mitochondria in eukaryotes, it is relevant to evaluate the fractional area that would be occupied were they to be located in the cell membrane. Such hypothetical packing densities are 5.0% and 6.6%, respectively, for the yeasts *S. cerevisiae* and *S. pombe*, and 6.6% and 6.8% for mouse fibroblasts and human HeLa cells. Although these observations suggest a ~5 fold increase in ATP synthase abundance with cell surface area in eukaryotes, the data conform to a continuous allometric function with no dichotomous break between the bacteria and eukaryotes (*Figure 1b*).

Similar conclusions can be reached regarding the ETCs, although direct comparisons are more difficult due to the diversity of electron transport chain complexes in prokaryotes (*Price and Driessen, 2010*). The number of ETC complexes is comparable to that of ATP synthases in both bacteria and eukaryotes (Supplementary Material), and the physical footprint of the ETC is ~5× that of $F_0F_1$ (~570 nm$^2$; *Dudkina et al., 2011*), implying that an average of ~5.5% of bacterial cell membranes is dedicated to the ETC and that the corresponding hypothetical packing density for eukaryotes would be ~30% (if in the cell membrane).

There are a number of uncertainties in these packing-density estimates, and more direct estimates are desirable. The optimum and maximum-possible packing densities for ATP synthase also remain unclear. Nonetheless, the fact remains that any packing problems that exist for the cell membrane are also relevant to mitochondrial membranes, which have additional protein components (such as those involved in internal-membrane folding and transport into and out of the mitochondrion).

## The biosynthetic cost of lipids

Any attempt to determine the implications of membranes for cellular evolution must account for the high biosynthetic costs of lipid molecules. There are two ways to quantify such a cost. First, from an evolutionary perspective, the cost of synthesizing a molecule is taken to be the sum of the direct use of ATP in the biosynthetic pathway plus the indirect loss of ATP resulting from the use of metabolic precursors that would otherwise be converted to ATP and available for alternative cellular functions (*Akashi and Gojobori, 2002*; *Lynch and Marinov, 2015*). Second, to simply quantify the direct contribution to a cell's total ATP requirement, the costs of diverting metabolic precursors are ignored.

By summing the total costs of all molecules underlying a cellular feature and scaling by the lifetime energy expenditure of the cell, one obtains a measure of the relative drain on the cell's energy budget associated with building and maintaining the trait. This measurement, $s_c$, can then be viewed as the fractional increase in the cell's energy budget that could be allocated to growth, reproduction, and survival in the absence of such an investment, ignoring the direct fitness benefits of expressing the trait, $s_a$. For selection to be effective, the net selective advantage of the trait, $s_n = s_a - s_c$, must exceed the power of random genetic drift, $1/N_e$ in a haploid species and $1/(2N_e)$ in a diploid, where $N_e$ is the effective population size.

Most cellular membranes are predominantly comprised of glycerophospholipids, which despite containing a variety of head groups (e.g. glycerol, choline, serine, and inositol), all have total biosynthetic costs per molecule (in units of ATP hydrolyses, and including the cost of diverting intermediate metabolites) of

$$c_L \simeq 320 + [38 \cdot (N_L - 16)] + (6 \cdot N_U), \tag{2a}$$
$$c_L \simeq 340 + [40 \cdot (N_L - 16)] + (6 \cdot N_U), \tag{2b}$$

in bacteria and eukaryotes, respectively, where $N_L$ is the mean fatty-acid chain length, and $N_U$ is the mean number of unsaturated carbons per fatty-acid chain (Supplementary material). Although variants on glycerophospholipids are utilized in a variety of species (*Guschina and Harwood, 2006*; *Geiger et al., 2010*), these are structurally similar enough that the preceding expressions should still provide excellent first-order approximations. The reduced (direct) cost, which ignores the loss of ATP-generating potential from the diversion of metabolic precursors, is

$$c'_L \simeq 110 + [7 \cdot (N_L - 16)] + (6 \cdot N_U), \tag{3a}$$

$$c'_L \simeq 120 + [9 \cdot (N_L - 16)] + (6 \cdot N_U), \tag{3b}$$

in bacteria and eukaryotes, respectively. From the standpoint of a cell's total energy budget, the evolutionary cost of a lipid molecule is $c_L/C_T$.

For most lipids in biological membranes, $14 \leq N_L \leq 22$ and $0 \leq N_U \leq 6$, so the cost per lipid molecule is generally in the range of $c_L \simeq 200$ to 600 ATP, although the average over all lipids deployed in species-specific membranes is much narrower (below). Cardiolipin, which rarely constitutes more than 20% of membrane lipids is exceptional, having an evolutionary cost of ~640 ATP/molecule (and a reduced cost of ~240 ATP). To put these expenditures into perspective, the evolutionary biosynthetic costs of each of the four nucleotides is $\approx$ 50 ATP hydrolyses per molecule (*Lynch and Marinov, 2015*), whereas the average cost of an amino acid is $\approx$ 30 ATP (*Atkinson, 1970*; *Akashi and Gojobori, 2002*).

Application of the preceding expressions to the known membrane compositions of cells indicates that the biosynthetic costs of eukaryotic lipids are higher than those in bacteria (Supplementary table). For example, for a diversity of bacterial species the average direct cost per lipid molecule in the plasma membrane is 123 (SE = 3) ATP, whereas that for eukaryotes is 143 (2). The latter estimate is identical to the mean obtained for whole eukaryotic cells, but the cost of mitochondrial lipids is especially high, 155 (5). These elevated expenses in eukaryotes are joint effects of the cost of mitochondrial export of oxaloacetate to generate acetyl-CoA and the tendency for eukaryotic lipids to have longer chains containing more desaturated carbons.

To understand the total bioenergetic cost associated with membranes, we require information on the numbers of lipid molecules required for membrane formation, which is equivalent to the total surface area of the membrane divided by the number of lipid molecules/unit surface area, and multiplied by two to account for the lipid bilayer. Estimates of the head-group areas of membrane lipids are mostly within 10% of an average value of $6.5 \times 10^{-7}$ $\mu m^2$ (*Petrache et al., 2000*; *Kučerka et al., 2011*), so the cost of a membrane (in units of ATP, and ignoring lipid turnover and the space occupied by transmembrane proteins) is

$$C_L \simeq (3.1 \times 10^6) \cdot \overline{c}_L \cdot A, \tag{4}$$

where $A$ is the membrane surface area in units of $\mu m^2$, and $\overline{c}_L$ is the average cost of a lipid.

Enough information is available on the total investment in mitochondrial membranes that a general statement can be made. Over the eukaryotic domain, the total surface area of mitochondria (inner plus outer membranes, summed over all mitochondria, in $\mu m^2$) scales with cell volume ($V$, in units of $\mu m^3$) as $3.0V^{0.99}$ (*Figure 1c*; SEs of intercept and slope on log plots are 0.22 and 0.08, respectively). Applying this to *Equation (4)*, with the average total cost of mitochondrial lipids ($\overline{c}_L = 440$ ATP/ molecule; *Appendix 1–table 4*), and using the expression for the total growth requirements of a cell, *Equation (1b)*, the relative cost of mitochondrial membrane lipids is

$$s_c \simeq 0.15V^{0.02}, \tag{5}$$

or ~15% of the total growth budget of a minimum-sized (1 $\mu m^3$) eukaryotic cell, and nearly independent of cell size within the range typically found in eukaryotes (SE of the exponent is 0.08). The direct contribution of mitochondrial membrane lipids to a cell's growth budget is ~36% of this total cost. These costs of mitochondrial membranes represent a baseline price, not incurred by prokaryotes, associated with relocating bioenergetics to the interior of eukaryotic cells, that is, ~5%. Unfortunately, the additional costs of maintenance of mitochondrial lipids is unknown, but for rapidly growing cells, the vast majority of a cell's energy budget is allocated to growth (*Lynch and Marinov, 2015*), so the above costs should still apply as first-order approximations; for slowly growing cells, the costs will be higher or lower depending on whether the cost of mitochondrial-membrane maintenance is above or below that for total cellular maintenance. Proteins do not occupy >50% of membranes, so accounting for this would change the preceding results by a factor <2.

For prokaryotic cells without internal membranes, the relative contribution of the cell membrane to a cell's total energy budget is expected to decline with increasing cell size, owing to the decline in the surface area to volume ratio. For the tiny cells of *Leptospira interrogans* and *Mycoplasma*

*pneumoniae* (average volumes of 0.03 and 0.22 $\mu m^3$, respectively), ~63 and 43% of a cell's growth budget must be allocated to the plasma membrane, but for the larger *Bacillus subtilis* and *Escherichia coli* (on average, 1.4 and 1.0 $\mu m^3$, respectively), these contributions drop to ~14 and 19%, and they would be expected to continue to decline with further increases in cell size, scaling inversely with the linear dimension of the cell.

In contrast, owing to the increased investment in internal membranes, the fraction of a eukaryotic cell's energy budget devoted to membranes does not diminish with increasing cell size. Although there are only a few eukaryotic cell types for which this issue can be evaluated quantitatively (*Table 1*), the data span three orders of magnitude in cell volume and uniformly suggest that ~10 to 30% of the total growth budget is allocated to lipid biosynthesis, and that an increasing fraction of such costs is associated with internal membranes in cells of increasing size. The picoplanktonic alga *Ostreococcus*, which has a cell volume of just 0.22 $\mu m^3$ (below that of many prokaryotes), devotes ~32% of its energy budget to membranes, and 44% of these costs (~18% of the total cell budget) are associated with internal membranes. A moderate-sized mammalian cell devotes a similar ~30% of its energy budget to membranes, but 96% of these costs (~29% of the total cell budget) are associated with internal membranes.

Taken together, these observations imply that the use of internal membranes constitutes a major drain on the total energy budgets of eukaryotic cells, much more than would be expected in bacteria of comparable size. Moreover, because the lipids associated with mitochondria alone constitute 20% to 35% of a eukaryotic cell's investment in membranes (*Table 1*), the energetic burden of localizing membrane bioenergetics to mitochondria is substantial.

Finally, given that the observations summarized in *Figure 1a,b* are derived from a diversity of studies, likely with many unique inaccuracies, it is worth considering whether the overall conclusions are consistent with the known capacity of ATP synthase. First, it bears noting that only a fraction of the energy invested in biosynthesis is derived directly from the chemiosmotic activity of ATP synthase. For example, amino-acid biosynthesis involves ~1.5 oxidations of NADH and NADPH for every ATP hydrolysis (*Akashi and Gojobori, 2002*). Assuming that each of the former is equivalent to ~3 ATP hydrolyses, this implies that only ~18% of the energy invested in amino-acid biosynthesis involves ATP hydrolysis. As noted in the Supplementary text, the ratio of use of NADH/NADPH to ATP is more on the order of 2.0 in lipid biosynthesis, reducing the direct investment in ATP to ~14% Thus, as the vast majority of the energetic cost of building a cell is associated with synthesis of the

**Table 1.** Contributions of membranes to total cellular growth costs.

Ot denotes the green alga *Ostreococcus tauri*, Sc the yeast *Saccharomyces cerevisiae*, Ds the green alga *Dunaliella salina*, and Ss the pig (*Sus scrofa*) pancreas cell; references given in Supplementary material. Cell volumes and total membrane areas are in units of $\mu m^3$ and $\mu m^2$, respectively, with the latter excluding membranes associated with the plastids in the algal species. The fraction of the total cell growth budget allocated to membranes is obtained by the ratio of Equations (1b) and (4), using the species-specific reduced costs in *Table 1* where available, and otherwise applying the averages for eukaryotic species; this total cost is then apportioned into five different fractional contributions in the following lines.

|  | Ot | Sc | Ds | Ss |
|---|---|---|---|---|
| Cell volume | 1 | 44 | 591 | 1060 |
| Total membranes | 15 | 204 | 2299 | 12952 |
| Fraction of absolute cell growth budget | 0.324 | 0.096 | 0.094 | 0.302 |
| Plasma membrane | 0.556 | 0.328 | 0.134 | 0.044 |
| Mitochondria | 0.243 | 0.359 | 0.197 | 0.223 |
| Nucleus | 0.113 | 0.085 | 0.034 | 0.008 |
| Endoplasmic reticulum + Golgi | 0.034 | 0.111 | 0.318 | 0.706 |
| Vesicles/vacuoles | 0.055 | 0.114 | 0.316 | 0.019 |

DOI: https://doi.org/10.7554/eLife.20437.004

monomeric building blocks of proteins and membranes, only ~15% of biosynthetic energy may be derived from ATP hydrolysis.

Given the known energy requirements for the maintenance and growth of a cell, the cell-division time, and the number of ATP synthase complexes per cell, it is possible to estimate the required rate of ADP → ATP conversions per complex. Using the cellular energetic data previously presented (*Lynch and Marinov, 2015*) and the ATP synthase abundances in *Appendix 1–table 2*, after discounting the maximum values by 85%, the estimated required rates of ATP production/complex/sec are: 2109, 221, and 19 respectively for the bacteria *B. subtilis*, *E. coli*, and *M. pneumoniae*, and 1440 and 329 for the yeasts *S. cerevisiae* and *S. pombe*. Several attempts have been made to estimate the maximum turnover rates (per sec) for $F_0F_1$ ATP synthase, usually in reconstituted liposomes, and these average 195/s in bacteria (*Etzold et al., 1997*; *Slooten and Vandenbranden, 1989*; *Toei et al., 2007*), 295 in soybean plastids (*Schmidt and Gräbe, 1985*; *Junesch and Gräber, 1991*), 120 in *S. cerevisiae* (*Förster et al., 2010*), and 440 in bovine heart (*Matsuno-Yagi and Hatefi, 1988*). Thus, given that a substantial fraction of complexes are likely to be misassembled in artificial membranes, the energy-budget based estimates of the numbers of ATP turnovers generated per cell appear to be consistent with the known capacity of ATP synthase.

## The cellular investment in ribosomes

The ribosome content of a cell provides a strong indicator of its bioenergetic capacity. Owing to the large number of proteins required to build the complex, ribosomes are energetically costly, and the number per cell appears to be universally correlated with cellular growth rate (*Fraenkel and Neidhardt, 1961*; *Tempest et al., 1965*; *Brown and Rose, 1969*; *Poyton, 1973*; *Dennis and Bremer, 1974*; *Freyssinet and Schiff, 1974*; *Alberghina et al., 1975*; *Boehlke and Friesen, 1975*; *Waldron and Lacroute, 1975*; *Scott et al., 2010*).

We previously pointed out that the genome-wide total and mean number of transcripts per gene scale with cell volume as $V^{0.36}$ and $V^{0.28}$ respectively, and that the analogous scalings are $V^{0.93}$ and $V^{0.66}$ for proteins, with no dichotomous break between prokaryotes and eukaryotes (*Lynch and Marinov, 2015*). As with the transcripts they process and the proteins they produce, the numbers of ribosomes per cell also appear to scale sublinearly with cell volume, in a continuous fashion across bacteria, unicellular eukaryotes, and cells derived from multicellular species (*Figure 2*). These observations are inconsistent with the idea that entry into the eukaryotic world resulted in an elevated rate of protein production. Moreover, as noted previously (*Lynch and Marinov, 2015*), the absolute costs of producing individual proteins and maintaining the genes associated with them are substantially higher in eukaryotes than in bacteria, owing to the substantial increase in gene lengths, investment in nucleosomes, etc.

## Discussion

*Lane (2015)* and *Lane and Martin (2010)* have proposed a scenario for how the mitochondrion became established by a series of adaptive steps, arguing that the eukaryotic leap to increased gene number and cellular complexity, and a subsequent adaptive cascade of morphological diversification, 'was strictly dependent on mitochondrial power'. However, the scaling of the costs of building and maintaining cells is inconsistent with an abrupt shift in volumetric bioenergetic capacity of eukaryotic cells, and although the absolute costs of biosynthesis, maintenance, and operation of individual genes are much greater in eukaryotes, the proportional costs are less (*Lynch and Marinov, 2015*). This means that evolutionary additions and modifications of genes are more easily accrued in eukaryotic genomes from a bioenergetics perspective, regardless of their downstream fitness effects.

The analyses presented here reveal a number of additional scaling features involving cellular bioenergetic capacity that appear to transcend the substantial morphological differences across the bacterial-eukaryotic divide. There is not a quantum leap in the surface area of bioenergetic membranes exploited in eukaryotes relative to what would be possible on the cell surface alone, nor is the idea that ATP synthesis is limited by total membrane surface area supported. Moreover, the numbers of both ribosomes and ATP synthase complexes per cell, which jointly serve as indicators of a cell's capacity to convert energy into biomass, scale with cell size in a continuous fashion both within and between bacterial and eukaryotic groups. Although there is considerable room for further

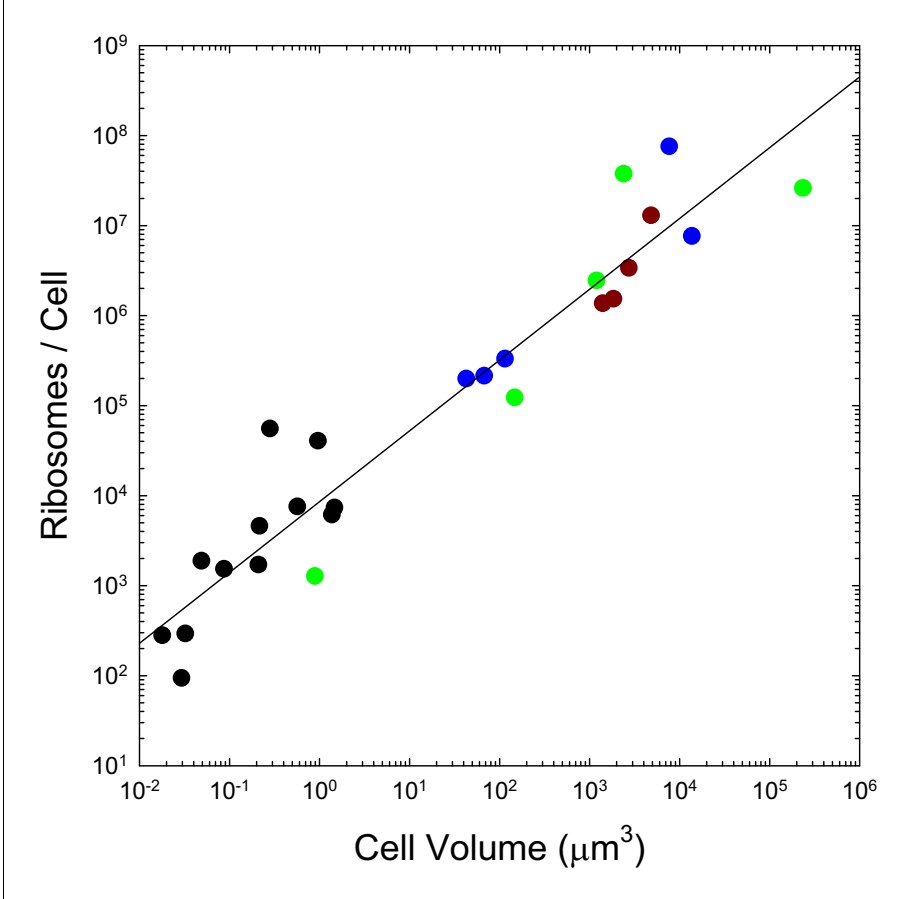

**Figure 2.** The number of ribosomes per cell scales with cell volume (V, in $\mu m^3$) as $7586V^{0.82}$ ($r^2$ = 0.92; SEs of the intercept and slope on the log scale are 0.13 and 0.05, respectively). Color coding as in previous figures. The data presented in this figure can be found in *Figure 2—source data 1*; see also *Appendix 1–table 3*.

DOI: https://doi.org/10.7554/eLife.20437.005

The following source data is available for figure 2:

**Source data 1.** Source data for *Figure 2*.

DOI: https://doi.org/10.7554/eLife.20437.006

comparative analyses in this area, when one additionally considers the substantial cost of building mitochondria, it is difficult to accept the idea that the establishment of the mitochondrion led to a major advance in net bioenergetic capacity.

Most discussion of the origin of the mitochondrion by endosymbiosis starts (and often ends) with a consideration of the benefits gained by the host cell. This ignores the fact that the eukaryotic consortium consists of two participants. At least initially, the establishment of a stable symbiotic relationship requires that each member of the pair gain as much from the association as is lost by relinquishing independence. Under the scenario painted by *Lane and Martin (2010)*, and earlier by *Martin and Müller (1998)*, the original mitochondrial-host cell affiliation was one in which the intracellular occupant provided hydrogen by-product to fuel methanogenesis in the host cell, while eventually giving up access to external resources and thereby coming to rely entirely on the host cell for organic substrates. For such a consortium to be evolutionarily stable as a true mutualism, both partners would have to acquire more resources than would be possible by living alone, in which case this novel relationship would be more than the sum of its parts.

Although some scenario like this might have existed in the earliest stages of mitochondrial establishment, it is also possible that one member of the original consortium was a parasite rather than a benevolent partner (made plausible by the fact that many of the $\alpha$-protobacteria to which

mitochondria are most closely related are intracellular parasites). Despite its disadvantages, such a system could be rendered stable if one member of the pair (the primordial mitochondrion) experienced relocation of just a single self-essential gene to the other member's genome, while the other lost a key function that was complemented by the presence of the endosymbiont. This scenario certainly applies today, as all mitochondria have relinquished virtually all genes for biosynthesis, replication, and maintenance, and as a consequence depend entirely on their host cells for these essential metabolic functions. In contrast, all eukaryotes have relocated membrane bioenergetics from the cell surface to mitochondrial membranes.

Such an outcome represents a possible grand example of the preservation of two ancestral components by complementary degenerative mutations (*Force et al., 1999*). Notably, this process of subfunctionalization is most likely to proceed in relatively small populations because the end state is slightly deleterious from the standpoint of mutational vulnerability, owing to the fact that the original set of tasks becomes reliant on a larger set of genes (*Lynch et al., 2001*). Thus, a plausible scenario is that the full eukaryotic cell plan emerged at least in part by initially nonadaptive processes made possible by a very strong and prolonged population bottleneck (*Lynch, 2007*; *Koonin, 2015*).

The origin of the mitochondrion was a singular event, and we may never know with certainty the early mechanisms involved in its establishment, nor the order of prior or subsequent events in the establishment of other eukaryotic cellular features (*Koonin, 2015*). However, the preceding observations suggest that if there was an energetic boost associated with the earliest stages of mitochondrial colonization, this has subsequently been offset by the loss of the use of the eukaryotic cell surface for bioenergetics and the resultant increase in costs associated with the construction of internal membranes. Rather than a major bioenergetic revolution being provoked by the origin of the mitochondrion, at best a zero-sum game is implied.

Thus, if the establishment of the mitochondrion was a key innovation in the adaptive radiation of eukaryotes, the causal connection does not appear to involve a boost in energy acquisition. Notably, a recent analysis suggests that the origin of the mitochondrion postdated the establishment of many aspects of eukaryotic cellular complexity (*Pittis and Gabaldón, 2016*). It is plausible, that phagocytosis was a late-comer in this series of events, made possible only after the movement of membrane bioenergetics to the mitochondrion, which would have eliminated the presumably disruptive effects of ingesting surface membranes containing the ETC and ATP synthase.

## Materials and methods

The results in this paper are derived from an integration of bioenergetic analyses based on known biochemical pathways and existing morphological observations on a variety of cell-biological features. The sources of this information, as well as the basic approaches employed can be found in the Appendix (where not mentioned directly in the text). The central analyses involve: (1) estimation of the biosynthetic costs for lipid-molecule production (in terms of ATP equivalents per molecule produced); (2) mitochondrial surface areas and cell membrane areas; (3) investments in lipids at the cell-membrane and organelle levels; and (4) numbers of ATP synthase complexes, ETC complexes, and ribosomes per cell.

## Acknowledgements

Support was provided by the Multidisciplinary University Research Initiative awards W911NF-09-1-0444 and W911NF-14-1-0411 from the US Army Research Office, National Institutes of Health award R01-GM036827, and National Science Foundation award MCB-1050161. This material is also based upon work supported by the National Science Foundation grant CNS-0521433. We are grateful to J Dacks, D Devos, J McKinlay, J Murray, and R Phillips for helpful comments.

# Additional information

## Funding

| Funder | Grant reference number | Author |
| --- | --- | --- |
| National Science Foundation | MCB-1050161 | Michael Lynch<br>Georgi K Marinov |
| National Institute of General Medical Sciences | R01-GM036827 | Michael Lynch<br>Georgi K Marinov |
| US Army Research Office | | Michael Lynch |
| US Army Research Office | W911NF-14-1-0411 | Michael Lynch |

The funders had no role in study design, data collection and interpretation, or the decision to submit the work for publication.

## Author contributions

Michael Lynch, Conceptualization, Data curation, Funding acquisition, Validation, Investigation, Methodology, Writing—original draft, Project administration, Writing—review and editing; Georgi K Marinov, Data curation, Formal analysis, Investigation, Methodology, Writing—original draft

## Author ORCIDs

Michael Lynch http://orcid.org/0000-0002-1653-0642
Georgi K Marinov http://orcid.org/0000-0003-1822-7273

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

## Appendix

DOI: https://doi.org/10.7554/eLife.20437.007

### The biosynthetic costs of lipid molecules

The vast majority of lipids in most membranes are phospholipids, with a polar (hydrophilic) head group attached to a negatively charged phosphate, which in turn is attached to a glycerol-3-phosphate (G3P), which links to two fatty-acid chains. Diversity within this lipid family is associated with variation in: the nature of the head groups; the number of carbon atoms in the fatty-acid chains; and the number of double bonds connecting such carbon atoms (their presence leading to 'unsaturated' fatty acids). Common head groups are choline, ethanolamine, serine, glycerol, inositol, and phosphatidyl glycerol. In both bacteria and eukaryotes, fatty-acid chains usually contain 12 to 22 carbons, and only rarely harbor more than three unsaturated bonds.

Evaluation of the total cost of synthesizing a lipid molecule requires a separate consideration of the investments in the three molecular subcomponents: the fatty-acid tails; head groups; and linkers. As adhered to in **Lynch and Marinov (2015)**, such costs will be quantified in units of ATP usage, specifically relying on the number of phosphorus atoms released via hydrolyses of ATP molecules, the primary source of energy in most endergonic cellular reactions. CTP, which is utilized in a few reaction steps in lipid biosynthesis, will be treated as equivalent to ATP, and electron transfers resulting from conversions of NADH to $NAD^+$, NADPH to $NADP^+$, and $FADH_2$ to FAD will be taken to be equivalent to 3, 3, and 2 ATP hydrolyses, respectively (all conventions in biochemistry based on energetic analyses; it is assumed that $NADP^+$/NADPH is efficiently recycled and obtained from sources other than action of the NADH kinase, which would elevate the cost to four high-energy phosphate groups). The following results are derived from observations cataloged in most biochemistry text books:

- The starting point for the synthesis of most fatty acids is the production of one particular linear chain, palmitate, which contains 16 carbon atoms. Synthesis of this molecule takes place within a large complex, known as fatty-acid synthase. In bacteria, biosynthesis of each palmitate molecule requires the consumption of 8 acetyl-CoA molecules, 7 ATPs, and reductions of 14 NADPHs. Each molecule of acetyl-CoA is generally derived from a pyruvate molecule, but each acetyl-CoA molecule diverted to lipid production deprives the cell of one rotation of the energy producing citric-acid cycle, which would otherwise yield 3 NADH, 1 $FADH_2$, and 1 ATP per rotation; this leads to a net loss to the cell of the equivalent of 12 ATPs per acetyl-CoA molecule. Thus, the total cost of production of one molecule of palmitate in bacteria is $(8 \times 12) + (7 \times 1) + (14 \times 3) = 145$ ATP.
  Fatty-acid production is slightly more expensive in nonphotosynthetic eukaryotes, where acetyl-CoA is produced in the mitochondrion and reacts with oxaloacetate to produce citrate, which must then be exported. Cleavage of oxaloacetate in the cytosol regenerates acetyl-CoA at the expense of 1 ATP, and a series of reactions serve to return oxaloacetate to the citric-acid cycle in an effectively ATP neutral way. Thus, the cost of palmitate increases to $145 + 8 = 153$ ATP.
- Each additional pair of carbons added to the palmitate chain requires one additional acetyl-CoA, one additional ATP, and two additional NADPHs, or an equivalent of 19 ATPs in bacteria, and accounting for mitochondrial export increases this to 20 in eukaryotes.
- Each subsequent desaturation of a fatty-acid bond consumes one NADPH, or 3 ATP equivalents.
- The G3P linker emerges from one of the last steps in glycolysis, and its diversion to lipid production deprives the cell of one further step of ATP production as well as a subsequent pyruvate molecule. As pyruvate normally can yield the equivalent of 3 ATPs in the conversion to acetyl-CoA, which then would generate a net 12 ATPs following entry into the citric-acid cycle, the use of G3P as a linker in a lipid molecule has a cost of $1 + 3 + 12 = 16$ ATP. Linking each fatty-acid tail requires 1 ATP, and linking the head group involves two CTP hydrolyses.

- All that remains now is to add in the cost of synthesis of the head group, which we do here still assuming 16 saturated bonds in each fatty acid. In the case of phosphatidylglycerol, the head group is G3P, the cost of which is 16 ATP as just noted, so the total cost of this molecule in a bacterium is $\simeq (2 \cdot 145) + (16 + 4) + 16 = 326$ ATP. From **Akashi and Gojobori (2002)**, the cost of a serine is 10 ATP, so the total cost of a phosphatidylserine is 320 ATP, and because ethanolamine and choline are simple derivatives of serine, this closely approximates the costs of both phosphatidylethanolamine and phosphatidylcholine. The headgroup of phosphatidylinositol is inosital, which is derived from glucose-6-phosphate, diverting the latter from glycolysis and depriving the cell of the equivalent of 9 ATPs, so the total cost of production of this molecule is 319 ATP. As a first-order approximation, we will assume all of the above molecules to have a cost of 321 ATP when containing fully saturated fatty acids with chain length 16. Finally, cardiolipin is synthesized by the fusion of two phosphatidylglycerols and the release of one glycerol, so taking the return from the latter to be 15 ATP, the total cost per molecule produced is 637 ATP.

## Estimation of absolute protein copy numbers per cell

Information on absolute protein copy numbers per cell was collected from publicly available proteomics measurements (**Lu et al., 2007**; **Wiśniewski et al., 2012, 2014**; **Maass et al., 2011**; **Maier et al., 2011**; **Schmidt et al., 2011**; **Beck et al., 2009**; **Kulak et al., 2014**; **Ghaemmaghami et al., 2003**; **Marguerat et al., 2012**; **Schwanhäusser et al., 2011**) as well as from ribosome profiling data (as described in **Lynch and Marinov, 2015**).

The number of protein complexes $N_{PC}$ was calculated as follows:

$$N_{PC,raw} = \frac{\sum_p N_p / s_p}{|p|}$$

where $N_p$ are the estimated per cell copy numbers for each subunit $p$ with a stoichiometric ratio $s_p$. Clear outliers (i.e., subunits with zero or near-zero counts) were removed from the calculation.

As proteomics measurements may not be absolutely reliable, the raw estimates $N_{PC,raw}$ were then further corrected where possible by taking advantage of the availability of direct counts of the number of ribosomes per cell:

$$N_{PC,corr} = N_{PC,raw} \cdot c_R$$

where the ribosomal correction factor $c_R$ is determined as follows:

$$c_R = \frac{N_{R,direct}}{N_{R,raw}}$$

where $N_{R,raw}$ refers to the estimated ribosome copy numbers derived as above, while $N_{R,direct}$ is obtained from direct measurements of ribosome copies per cell.

The composition of the *E. coli* $F_O$-particle is $1a{:}2b{:}10{-}12c$ while that of the $F_1$-particle is $3\alpha{:}3\beta{:}1\delta{:}1\gamma{:}1\epsilon$ (**Jonckheere et al., 2012**; **Capaldi et al., 2000**), where the individual subunits are encoded by the following genes:

| Subunit | Gene |
|---|---|
| $a$ | *atpB* |
| $b$ | *atpF* |
| $c$ | *atpE* |
| $\alpha$ | *atpA* |
| $\beta$ | *atpD* |

*continued on next page*

*continued*

| Subunit | Gene |
|---|---|
| $\gamma$ | *atpG* |
| $\delta$ | *atpH* |
| $\epsilon$ | *atpC* |

The same composition and stoichiometry was also assumed for other prokaryotes.

The composition of the yeast $F_1$-particle is $3\alpha{:}3\beta{:}1\delta{:}1\gamma{:}1\epsilon{:}1$OSCP (*Jonckheere et al., 2012*). The $F_O$-particle has 10 copies of subunit 9 (equivalent to $c$), and one copy each of subunits 6 (equivalent to $a$), 8, 4 (equivalent to $b$), $d$, $h$, $f$, $e$, $g$, $i$ and $k$, where the individual subunits are encoded by the following genes:

| Subunit | Gene |
|---|---|
| $\alpha$ | *ATP1* |
| $\beta$ | *ATP2* |
| $\gamma$ | *ATP3* |
| $\delta$ | *ATP16* |
| $\epsilon$ | *ATP15* |
| $a$ | *MT-ATP6* |
| 4 | *ATP4* |
| 9 | *ATP9* |
| 8 | *MT-ATP8* |
| $d$ | *ATP7* |
| $e$ | *ATP21* |
| $h$ | *ATP14* |
| $f$ | *ATP17* |
| $g$ | *ATP20* |
| $i$ | *ATP18* |
| $k$ | *ATP19* |
| OSCP | *ATP5* |

The composition of the mammalian $F_1$-particle is $3\alpha{:}3\beta{:}1\delta{:}1\gamma{:}1\epsilon{:}1$OSCP (*Jonckheere et al., 2012*). The $F_O$-particle has 8 copies of subunit $c$, and one copy each of subunits $a$, 8, $b$, $d$, $F_6$, $f$, $e$, and $g$, where the individual subunits are encoded by the following genes:

| Subunit | Gene |
|---|---|
| $\alpha$ | *ATP5A1* |
| $\beta$ | *ATP5B* |
| $\gamma$ | *ATP5C1* |
| $\delta$ | *ATP5D* |
| $\epsilon$ | *ATP5E* |
| $a$ | *MT-ATP6* |
| $b$ | *ATP5F1* |
| $c$ | *ATP5G1*<br>*ATP5G2*<br>*ATP5G3* |
| 8 | *MT-ATP8* |

*continued on next page*

*continued*

| Subunit | Gene |
|---|---|
| *d* | *ATP5H* |
| *e* | *ATP5I* |
| $F_6$ | *ATP5J* |
| *f* | *ATP5J2* |
| *g* | *ATP5L* |
| OSCP | *ATP5O* |

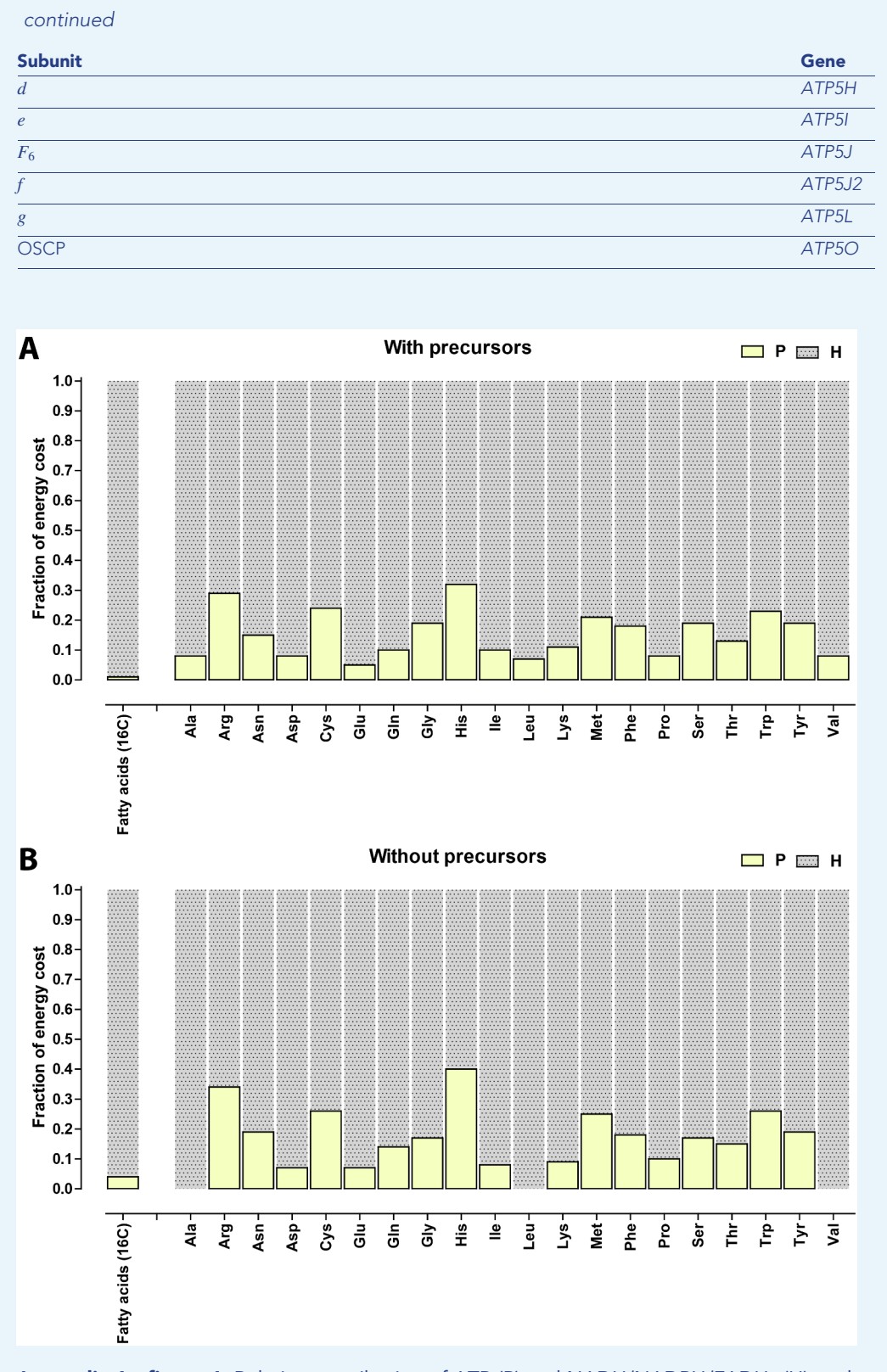

**Appendix 1—figure 1.** Relative contribution of ATP (P) and NADH/NADPH/FADH$_2$ (H) to the biosynthetic costs of lipids and amino acids. (**A**) Nonreduced costs including opportunity cost of precursors; (**B**) Reduced costs without precursors. Amino acid values are obtained from *Akashi and Gojobori (2002)*, assuming growth on glucose.

DOI: https://doi.org/10.7554/eLife.20437.011

**Appendix 1—table 1.** Features of mitochondrial membranes.
Cell volumes are from *Lynch and Marinov (2015)*, in some cases supplemented with additional references from the literature. $V$: cell volume (in μm³); $SA_C$: cellular surface area (in μm²); $SA_{MI}$: inner mitochondrial membrane surface area (in μm²); $SA_{MI+MO}$: inner+outer mitochondrial membrane surface area (in μm²); $MI/MO$ ratio between inner and outer mitochondrial membrane surface area

| Species | $V$ | $SA_C$ | $SA_{MI}$ | $SA_{MI+MO}$ | $MI/MO$ | References |
|---|---|---|---|---|---|---|
| Unicellular eukaryotes | | | | | | |
| Exophiala dermatitidis | 43.80 | 50.95 | 73.98 | | | Biswas et al. (2003) |
| Candida albicans | 35.36 | 96.10 | 37.37 | | | Tanaka et al. (1985); Klis et al. (2014) |
| Saccharomyces cerevisiae | 69.07 | 61.42 | 15.83 | | | Uchida et al. (2011) |
| Tetrahymena pyriformis | 16666.00 | 3014.05 | 12987.60 | 83968.50 | 5.200 | Gleason et al. (1975); Poole (1983) |
| Trichoderma viride | 126.70 | 122.01 | 139.40 | | | Rosen et al. (1974) |
| Mammals | | | | | | |
| Cat, gracilis muscle | | | | | 2.323 | Schwerzmann et al. (1989) |
| Hamster, intestinal enterocyte | 1890.00 | 5772.00 | 2668.00 | 9351.00 | 3.256 | Buschmann and Manke (1981a, 1981b) |
| Human HeLa cells | 2798.67 | 1178.00 | 1424.74 | | | |
| Mouse heart | | | | | 7.020 | Kistler and Weber (1975) |
| Mouse liver | | | | | 3.540 | Kistler and Weber (1975) |
| Mouse lymphocyte | 50.69 | 88.27 | 20.43 | | | Al-Hamdani et al. (1979); Mayhew et al. (1979) |
| Mouse immunoblast | 392.98 | 282.94 | 143.52 | | | Al-Hamdani et al. (1979) |
| Mouse pancreas | 1434.00 | 973.00 | 779.00 | | | Dean (1973) |
| Pig pancreas cell | 1060.00 | 581.90 | 460.50 | 2698.50 | 4.860 | Bolender (1974) |
| Rat Leydig cell, testes | 1210.00 | 1517.00 | 1641.00 | 4561.00 | 1.779 | Mori and Christensen (1980) |
| Rat liver cell | 5100.00 | 1680.00 | 7651.65 | 42615.56 | 4.718 | Weibel et al. (1969); Jakovcic et al. (1978) |
| Rat heart | | | | | 12.760 | Reith et al. (1973) |
| Rat L-8 skeletal muscle cell | | | | | 4.670 | Reith et al. (1973) |
| Land plants and algae | | | | | | |
| Arabidopsis thaliana, cotyledon | 5237.75 | | 1307.00 | | | |

Appendix 1—table 1 continued

| Species | V | $SA_C$ | $SA_{MI}$ | $SA_{MI+MO}$ | MI/MO | References |
|---|---|---|---|---|---|---|
| Chlamydomonas reinhardtii | 128.38 | 129.60 | 66.82 | | | Calvayrac et al. (1974); Hayashi and Ueda (1989) |
| Chlorella fusca | 102.00 | 111.40 | 48.40 | | | Atkinson et al. (1974); Forde et al. (1976) |
| Dunaliella salina | 590.80 | 322.50 | 87.40 | | | Maeda and Thompson (1986) |
| Medicago sativa, meristem | 166.90 | 221.50 | 16.00 | | | Zhu et al. (1991) |
| Ochromonas danica | | | | | 2.450 | Smith-Johannsen and Gibbs, 1972 |
| Ostreococcus tauri | 0.91 | 8.30 | 0.70 | | | Henderson et al. (2007) |
| Polytoma papillatum | 862.54 | 471.43 | 778.64 | | | Gaffal et al. (1982) |
| Rhus toxicodendron | 1222.00 | | | 1288.50 | 2.545 | Vassilyev (2000) |

DOI: https://doi.org/10.7554/eLife.20437.012

**Appendix 1—table 2.** Estimated abundance of ATP synthase complexes in species with quantitative proteomics data. ATP synthase surface area assumed to be maximum associated with the inner ring, $6.4 \times 10^{-5}$ m$^2$ for bacteria, $1.1 \times 10^{-4}$ for eukaryotes. $V$: cell volume (in µm$^3$); $SA_C$: cellular surface area (in µm$^2$); $N_{PC,raw}$: raw protein complex copy number estimates; $N_{PC,corr}$: corrected protein complex copy number estimates; $c_R$: correction factor; $PD$: packing density (copies/µm$^2$); $f_{SA}$: fraction of $SA$; $R_{max}$ and $R_{red}$: maximum (all ATP equivalents) and reduced (without ATP equivalents expended in the form of NADH/NADPH/FADH$_2$) required rate of ATP synthesis (per complex per second) to satisfy lifetime energy requirements.

| Species | $V$ | $SA_C$ | F$_0$F$_1$ copies per cell | | $c_R$ | $PD$ | $f_{SA}$ | $t$ | $C_G$ | $C_M$ | $C_T$ | $R_{max}$ | $R_{red}$ | References |
|---|---|---|---|---|---|---|---|---|---|---|---|---|---|---|
| | | | $N_{PC,raw}$ | $N_{PC,corr}$ | | | | | | | | | | |
| **Prokaryotes** | | | | | | | | | | | | | | |
| Bacillus subtilis | 1.407 | 10.69 | 2435 | 1602 | 0.66 | 150 | 0.010 | 1.16 | 92.51 | 1.16 | 93.85 | 14062 | 2109 | *Jeong et al. (1990); Weart et al. (2007); Sharpe et al. (1998)* |
| Escherichia coli | 0.983 | 10.85 | 1056 | 3018 | 2.86 | 278 | 0.018 | 0.99 | 15.65 | 0.21 | 15.86 | 1475 | 221 | *Young (2006); Milo and Phillips, 2016*  Beck et al. (2009) |
| Leptospira interrogans | 0.220 | 5.72 | 1187 | 1344 | NA | 235 | 0.015 | | | | | | | |
| Mycoplasma pneumoniae | 0.033 | 1.32 | 117 | 131 | 1.12 | 99 | 0.006 | 63.74 | 0.92 | 0.05 | 3.87 | 129 | 19 | *Zucker-Franklin et al. (1996a), 1996b* |
| Staphylococcus aureus | 0.288 | 4.00 | 447 | NA | NA | 112 | 0.007 | | | | | | | *Kehle and Herzog (1989)* |
| **Fungi** | | | | | | | | | | | | | | |
| Saccharomyces cerevisiae (hap) | 37.940 | 64.42 | 15659 | 29126 | 1.86 | 452 | 0.050 | 2.50 | 2468.20 | 18.79 | 2515.15 | 9598 | 1440 | |
| Schizosaccharomyces pombe | 118.000 | 116.38 | 65363 | 70129 | 1.07 | 603 | 0.066 | 4.31 | 2347.80 | 8.70 | 2385.29 | 2193 | 329 | |
| **Mammals** | | | | | | | | | | | | | | |
| Homo sapiens , HeLa cell | 2798.668 | 1178.00 | 1284376 | 737270 | 0.57 | 626 | 0.068 | | | | | | | *Borle (1969a, 1969b)* |
| Mus musculus , fibroblast NIH3T3 | 1765.000 | 2100.00 | 1255254 | NA | NA | 598 | 0.066 | | | | | | | *Schwanhäusser et al. (2011)* |

DOI: https://doi.org/10.7554/eLife.20437.013

**Appendix 1—table 3.** Estimated numbers of ribosomes per cell.
Direct estimates taken from microscopic examinations; proteomic estimates are from averaging of cell-specific estimates for each ribosomal protein subunit. $V$: cell volume (in μm³); $N_{R,direct}$: directly estimated copies per cell; $N_{R,raw}$: estimated copies per cell based on proteomics studies. See **Figure 2—source data 1** for further details.

| Species | $V$ | $N_{R,direct}$ | $N_{R,raw}$ | References |
|---|---|---|---|---|
| Bacteria | | | | |
| Bacillus subtilis | 1.44 | 6000 | | Barrera and Pan (2004) |
| | | | 9124 | Maass et al. (2011) |
| Escherichia coli | 0.93 | 72,000 | | Bremer and Dennis (1996) |
| | | 45,100 | | Fegatella et al. (1998) |
| | | 26,300 | | Fegatella et al. (1998) |
| | | 13,500 | | Fegatella et al. (1998) |
| | | 6800 | | Fegatella et al. (1998) |
| | | 55,000 | | Bakshi et al. (2012) |
| | | 20,100 | | |
| | | 12,000 | | Arfvidsson and Wahlund (2003) |
| | | | 6514 | Wiśniewski et al. (2014) |
| | | | 17,979 | Lu et al. (2007) |
| Legionella pneumophila | 0.58 | 7400 | | Leskelä et al. (2005) |
| Leptospira interrogans | 0.22 | 4500 | | Beck et al. (2009) |
| | | | 3745 | Schmidt et al. (2011) |
| Mycoplasma pneumonii | 0.05 | 140 | | Yus et al. (2009) |
| | | 300 | | Seybert et al. (2006) |
| | | 140 | | Kühner et al. (2009) |
| | | | 255 | Maier et al. (2011) |
| Mycobacterium tuberculosis | 0.21 | 1672 | | Yamada et al. (2015) |
| Rickettsia prowazekii | 0.09 | 1500 | | Pang and Winkler (1994) |

*Appendix 1—table 3 continued on next page*

**Appendix 1—table 3 continued**

| Species | | $V$ | $N_{R,direct}$ | $N_{R,raw}$ | References |
|---|---|---|---|---|---|
| Sphingopyxis alaskensis | | 0.07 | 1850 | | Fegatella et al. (1998) |
| | | | 200 | | Fegatella et al. (1998) |
| Spiroplasma melliferum | | 0.02 | 275 | | Ortiz et al. (2006) |
| Staphylococcus aureus | | 0.31 | 54,400 | | Martin and Iandolo (1975) |
| Vibrio angustum | | | 27,500 | | Flärdh et al. (1992) |
| | | | 8000 | | Flärdh et al. (1992) |
| Archaea | | | | | |
| ARMAN | undescribed | 0.03 | 92 | | Comolli et al. (2009) |
| Eukaryotes | | | | | |
| Exophiala dermatitidis | | 44 | 195,000 | | Biswas et al. (2003) |
| Saccharomyces cerevisiae | haploid | 68 | 200,000 | | Warner (1999) |
| | | | 220,000 | | Yamaguchi et al. (2011) |
| | | | | 134,438 | Kulak et al. (2014) |
| | | | | 74,800 | Ghaemmaghami et al. (2003) |
| Schizosaccharomyces pombe | | 133 | 150,000 | | Marguerat et al. (2012) |
| | | | 500,000 | | Maclean (1965) |
| | | | | 356,180 | Kulak et al. (2014) |
| | | | | 101,099 | Marguerat et al. (2012) |
| Tetrahymena pyriformis | | 14588 | 88,900,000 | | Hallberg and Bruns (1976) |
| Tetrahymena thermophila | | 12742 | 74,000,000 | | Calzone et al. (1983) |
| Chlamydomonas reinhardtii | cytoplasm | 139 | 120,500 | | Bourque et al. (1971) |
| | chloroplast | | 55,000 | | |
| Ostreococcus tauri | | 0.91 | 1250 | | Henderson et al. (2007) |
| Adonis aestivalis | vegetative | 2380 | 47,700,000 | | Lin and Gifford (1976) |

*Appendix 1—table 3 continued on next page*

*Appendix 1—table 3 continued*

| Species | | V | $N_{R,direct}$ | $N_{R,raw}$ | References |
|---|---|---|---|---|---|
| | transitional | 2287 | 39,066,666 | | |
| | floral | 2690 | 23,933,333 | | |
| Glycine max SB-1 cell | | | 9,373,333 | | *Jackson and Lark (1982)* |
| Rhus toxicodendron | | 1222 | 2,400,000 | | *Vassilyev (2000)* |
| Zea mays root cell | | 240,000 | 25,500,000 | | *Hsiao and (1970)* |
| Hamster, intestinal enterocyte | | 1890 | 1,500,000 | | *Buschmann and Manke (1981a, 1981b)* |
| HeLa cell | | 2800 | 3,150,000 | | *Duncan and Hershey (1983)* |
| | | | | | *Zhao et al. (2008)* |
| | | | | 4,631,143 | *Kulak et al. (2014)* |
| Mouse pancreas | | 1434 | 1,340,000 | | *Dean (1973)* |
| Rat liver cell | | 4940 | 12,700,000 | | *Weibel et al. (1969)* |

DOI: https://doi.org/10.7554/eLife.20437.014

**Appendix 1—table 4.** Costs of lipids. The average cost per molecule is calculated for a variety of species using estimates of lipid compositions from the literature and the formulas described in the text. The fraction of fatty acids of given length and saturation level is not shown. Cardiolipin costs are assumed to be 637 (evolutionary) and 236 (reduced) ATP. The cost for molecules in the 'other' category is assumed to be the average of glycerophospholipids (GPL) in the species and cardiolipin.

| Species | Membrane | GPL cost | | Composition | | | Mean cost | | | References |
|---|---|---|---|---|---|---|---|---|---|---|
| | | Tot. | Red. | GPL | Cardiolipin | Other | Tot. | Red. | | |
| Escherichia coli | Whole cell | 367 | 115 | 0.926 | 0.060 | 0.015 | 385 | 124 | | *Haest et al. (1969); Rietveld et al. (1993); Raetz et al. (1979)* |
| Bacillus subtilis | Whole cell | 308 | 102 | 0.818 | 0.183 | 0.000 | 368 | 127 | | *Bishop et al. (1967); López et al. (1998)* |
| Caulobacter crescentus | Whole cell | 340 | 111 | 0.776 | 0.105 | 0.119 | 389 | 132 | | *Contreras et al. (1978); Chow and Schmidt (1974)* |
| Staphylococcus aureus | Whole cell | 323 | 105 | 0.931 | 0.070 | 0.000 | 345 | 114 | | *Haest et al. (1972); Mishra and Bayer (2013)* |
| Zymomonas mobilis | Whole cell | 370 | 118 | 0.990 | 0.010 | 0.000 | 373 | 119 | | *Carey and Ingram (1983)* |
| | | | | | | | 372 | 123 | mean | |
| | | | | | | | 8 | 3 | SE | |
| Candida albicans | Whole cell | 338 | 123 | 0.934 | 0.066 | 0.000 | 358 | 131 | | *Goyal and Khuller (1992); Singh et al. (2010)* |
| Chlamydomonas reinhardtii | Whole cell | 390 | 140 | 0.935 | 0.065 | 0.000 | 406 | 146 | | *Janero and Barrnett (1981); Giroud et al. (1988); Tatsuzawa et al. (1996)* |
| Debaryomyces hansenii | Whole cell | 408 | 141 | 0.913 | 0.087 | 0.000 | 428 | 150 | | *Kaneko et al. (1976)* |
| Dictyostelium discoideum | Whole cell | 400 | 141 | 0.965 | 0.014 | 0.000 | 395 | 139 | | *Davidoff and Korn (1963); Ellingson (1974); Weeks and Herring (1980); Paquet et al. (2013)* |
| Paramecium tetraurelia | Whole cell | 415 | 146 | 0.996 | 0.004 | 0.000 | 415 | 146 | | |
| Pichia pastoris | Whole cell | 412 | 144 | 0.975 | 0.025 | 0.000 | 418 | 147 | | *Klug et al. (2014)* |
| Saccharomyces cerevisiae | Whole cell | 372 | 133 | 0.953 | 0.047 | 0.000 | 385 | 138 | | *Longley et al. (1968); Kaneko et al. (1976); Sharma (2006); Klis et al. (2014)* |
| Schizosaccharomyces pombe | Whole cell | 411 | 142 | 0.945 | 0.055 | 0.000 | 424 | 147 | | *Koukou et al. (1990)* |
| | | | | | | | 403 | 143 | mean | |
| | | | | | | | 8 | 2 | SE | |
| Debaryomyces hansenii | Plasma membrane | 398 | 137 | 0.913 | 0.087 | 0.000 | 418 | 146 | | *Kaneko et al. (1976); Turk et al. (2007)* |
| Dictyostelium discoideum | Plasma membrane | 414 | 145 | 0.980 | 0.020 | 0.000 | 418 | 147 | | *Weeks and Herring (1980)* |
| Dunaliella salina | Plasma membrane | 378 | 137 | 1.000 | 0.000 | 0.000 | 378 | 137 | | *Peeler et al. (1989); Azachi et al. (2002)* |
| Mus musculus , thymocytes | Plasma membrane | 409 | 142 | 0.921 | 0.000 | 0.079 | 418 | 145 | | *Van Blitterswijk et al. (1982)* |

*Appendix 1—table 4 continued on next page*

*Appendix 1—table 4 continued*

| Species | Membrane | GPL cost | | Composition | | | Mean cost | | | References |
|---|---|---|---|---|---|---|---|---|---|---|
| | | Tot. | Red. | GPL | Cardiolipin | Other | Tot. | Red. | | |
| *Saccharomyces cerevisiae* | Plasma membrane | 358 | 129 | 0.949 | 0.035 | 0.026 | 375 | 135 | | *Longley et al. (1968)*; *Zinser et al. (1991)*; *Swan and Watson (1997)*; *Tuller et al. (1999)*; *Blagović et al. (2005)* |
| *Schizosaccharomyces pombe* | Plasma membrane | 411 | 142 | 0.856 | 0.052 | 0.092 | 433 | 151 | | *Koukou et al. (1990)* |
| *Vigna radiata*, seedling | Plasma membrane | 402 | 141 | 1.000 | 0.000 | 0.000 | 402 | 141 | | *Yoshida and Uemura (1986)* |
| | | | | | | | 406 | 143 | mean | |
| | | | | | | | 8 | 2 | SE | |
| *Candida albicans* | Mitochondrion | 344 | 125 | 0.710 | 0.164 | 0.126 | 411 | 150 | | *Goyal and Khuller (1992)* |
| *Danio rerio*, whole fish | Mitochondrion | 472 | 162 | 0.854 | 0.104 | 0.042 | 492 | 172 | | *Almaida-Pagán et al. (2014)* |
| *Pichia pastoris* | Mitochondrion | 421 | 145 | 0.944 | 0.054 | 0.002 | 433 | 150 | | *Wriessnegger et al. (2009)*; *Klug et al. (2014)* |
| *Rattus norwegicus*, liver | Mitochondrion | 445 | 154 | 0.838 | 0.148 | 0.024 | 480 | 169 | | *Tahin et al. (1981)*; *Colbeau et al. (1971)* |
| *Saccharomyces cerevisiae* | Mitochondrion | 312 | 116 | 0.897 | 0.097 | 0.006 | 345 | 128 | | *Tuller et al. (1999)*; *Zinser et al. (1991)*; *Blagović et al. (2005)* |
| *Serripes groenlandicus*, gill | Mitochondrion | 428 | 147 | 0.972 | 0.028 | 0.000 | 434 | 150 | | *Gillis and Ballantyne (1999)* |
| *Sus scrofa*, heart | Mitochondrion | 409 | 143 | 0.797 | 0.186 | 0.017 | 453 | 161 | | *Comte et al. (1976)* |
| *Tetrahymena pyriformis* | Mitochondrion | 402 | 144 | 0.812 | 0.131 | 0.057 | 439 | 159 | | *Gleason (1976)*; *Nozawa (2011)* |
| | | | | | | | 436 | 155 | mean | |
| | | | | | | | 16 | 5 | SE | |

DOI: https://doi.org/10.7554/eLife.20437.015

