## [Decision Letter]

Thank you for submitting your article "Membranes, Energetics, and Evolution Across the Prokaryote-Eukaryote Divide" for consideration by *eLife*. Your article has been reviewed by two peer reviewers, and the evaluation has been overseen by Paul Falkowski as the Reviewing Editor and Patricia Wittkopp as the Senior Editor. The following individual involved in review of your submission has agreed to reveal his identity: Ron Milo (Reviewer #2).

The reviewers have discussed the reviews with one another and the Reviewing Editor has drafted this decision to help you prepare a revised submission.

Summary: Both reviewers identified many strengths of this work, but also have identified additional elements to consider. I hope you find their detailed and constructive reviews helpful. We anticipate that this work will be an important contribution to the field that will spark additional discussion and debate.

Essential revisions: Both reviewers have provided detailed reviews of this manuscript, and we believe that considering all of their comments will be beneficial in this case. These comments are provided in their entirety below. The most essential comment from the reviewers that must be addressed is: The "possibility that protein packing density in the membranes under consideration is a fundamental limitation needs to be taken into account."

*Reviewer #1:*

This is an interesting analysis of the relative bioenergetic characteristics in the growth of prokaryotic and eukaryotic cells. The article addresses the basic conjecture, that the evolution of the eukaryotic type, specifically the development of mitochondrial systems, endowed the eukaryotes with energetic advantages over the prokaryotic cellular organization. The authors are challenging this often-made, yet largely unsubstantiated, assumption that the presence of mitochondria in eukaryotes confers a large bioenergetic advantage owing to a corresponding increase in internal membrane surface area due to the presence of the mitochondrial inner membrane. To address this question, the authors perform an analysis based upon previous scaling relationships they have developed between quantities such as the volume of a cell and the rates of ATP consumption and combined these with a new analysis that includes protein and lipid abundances combined with estimations, from the literature, of their costs as expressed in terms of ATP equivalents.

The authors note that the energetics of the cell can be divided into maintenance costs and the costs of duplicating the parental cell and their analysis goes on from there. Basically, they are concluding that if there ever was an energetic advantage (e.g. on a cell volume basis), then it no longer exists and that the eukaryotic cell type does not confer energetic advantages. Overall, I think the article is sound, albeit, it is difficult for this reviewer to critically assess the validity of their calculations, which on the surface seems sound. On the other hand, the article is written in a with the tenor of a polemic and is a bit rambling. Consequently, I believe it needs to be considerably shortened (25%).

1) Subsection “The energetic costs of building and maintaining a cell”, second paragraph: authors should cite Daniel Atkinson on the biosynthetic costs.

2)Subsection “The energetic costs of building and maintaining a cell”, last paragraph: A relatively simple scaling relationship for bacterial growth may apply for certain species, but it needs to be pointed out that at either end of the range in size there are slowdowns in growth rate, with certain larger bacteria, for example, having more protracted cell division times.

3) The possibility that protein packing density in the membranes under consideration is a fundamental limitation needs to be taken into account. My recollection is that many membrane systems are at least 50% protein by weight. It may be true that the bioenergetic machinery responsible for ATP production only occupies several percent of the total area, but this may be the upper limit for the bioenergetic system reflecting and optimal allocation of different protein functions, such as transporters, also necessary for metabolism. Presumably, the other mitochondrial components especially are present in an optimal stoichiometric ratio with respect to the ATP synthase and may indeed occupy much more of the membrane area. For example, if the ATP synthase has an intrinsically higher enzymatic turnover frequency than the enzymes powering the generation of proton motive force, then it's amount can be comparatively small on a stoichiometric basis and the other membrane complexes may occupy a large fraction of the membrane surface.

*Reviewer #2:*

The authors revisit the hypothesis that the mitochondria were essential for the development of eukaryotic complexity for energetic reasons. The authors thoroughly analyze the ATP and other investments as performed by current eukaryotic cells and compare them to prokaryotes. They use empirical scaling laws to see if the observed changes are more than one would expect from simple scaling with cell volume. They find no strong evidence for a significant energetic benefit from mitochondria which leads them to cast doubt on high profile earlier reports.

I find the study scientifically sound and interesting. I have suggestions for improvement in terms of clarity and accuracy as given below.

Main text, third paragraph: "This implies that the mitochondrion-host cell consortium that became the primordial eukaryote did not precipitate a bioenergetics revolution."

In order to say it did not cause a bioenergetics revolution I need to have a definition of what is the definition such a revolution in as rigorous terms as possible. Either by the authors or by them repeating in detail a definition from previous authors.

Throughout the paper the scaling laws have no uncertainty ranges on their parameter values. This makes it hard to understand how predictive they are and should be corrected.

Subsection “The energetic costs of building and maintaining a cell”, fourth paragraph: "that a shift of bioenergetics from the cell membrane in prokaryotes to the mitochondria of eukaryotes conferred no directly favorable energetic effects. In fact, the effect appears to be negative."

One could claim that because prokaryotic ATP production is associated with the cell membrane and it scales like the surface area an exponent of 1 with cell volume is not what one would expect (but rather 2/3) and the evidence supporting an approx ~1 exponent suggests there is some favorable energetic effect. I am not saying this is proof of such an effect but I think this point should be acknowledged/discussed.

Subsection “Energy production and the mitochondrion”, last sentence: "and that the corresponding hypothetical packing density for eukaryotes would be 30% (if in the cell membrane)."

The authors do not seem to reflect more on this value they derive but it seems like a very high value to me. Given that packing of equally sized circles on a sphere cannot achieve more than I think about 60% usage of the sphere area this is not far from the maximal possible and this is before considering all the other protein machines needed in the membrane real estate or the requirements for lipids.

Subsection “The biosynthetic cost of lipids”: "and *Escherichia coli* (… 0.98 μm^3^, respectively)"

The volume of an *E. coli* cell can easily change by a factor of 5 depending on growth rate so giving the volume as 0.98 μm^3^ without stating anything about growth conditions is odd. Better state as ~1 μm^3^ or the like.

Discussion, fifth paragraph: "because the end state is slightly deleterious owing to the additional investment required to carry out individual tasks (Lynch et al. 2001)."

I found it hard to follow the logic here and I think other readers might have this problem. It is worth explaining in a bit more detail what is meant.

Discussion, last paragraph: "It is plausible, that phagocytosis was a late-comer in this series of events, made possible only after the movement of membrane bioenergetics to the mitochondrion, which would have eliminated the disruptive effects of surface membrane ingestion on the ETC and ATP synthase."

I did not understand the connection here. Please clarify.

---

## [Author Response]

*Essential revisions: Both reviewers have provided detailed reviews of this manuscript, and we believe that considering all of their comments will be beneficial in this case. These comments are provided in their entirety below. The most essential comment from the reviewers that must be addressed is: The "possibility that protein packing density in the membranes under consideration is a fundamental limitation needs to be taken into account."*

*Reviewer #1:*

*[…] 1) Subsection “The energetic costs of building and maintaining a cell”, second paragraph: authors should cite Daniel Atkinson on the biosynthetic costs.*

Thank you for pointing this out; done. Fully admit to not having read this before, and it is remarkable how similar his results are to those of Akashi and Gojobori. Although he did not deal with lipids to any great extent, the little he did seems to be compatible with our calculations, so that is gratifying as well.

*2)Subsection “The energetic costs of building and maintaining a cell”, last paragraph: A relatively simple scaling relationship for bacterial growth may apply for certain species, but it needs to be pointed out that at either end of the range in size there are slowdowns in growth rate, with certain larger bacteria, for example, having more protracted cell division times.*

Our point is already that there is a slowdown in the growth rate of bacterial cells at the low end of the size range. We are less clear as to what species the reviewer is referring to at the large end, as we attempted to perform as thorough and as unbiased a survey as possible; we have emphasized that there if a broad range around the general pattern.

*3) The possibility that protein packing density in the membranes under consideration is a fundamental limitation needs to be taken into account. My recollection is that many membrane systems are at least 50% protein by weight. It may be true that the bioenergetic machinery responsible for ATP production only occupies several percent of the total area, but this may be the upper limit for the bioenergetic system reflecting and optimal allocation of different protein functions, such as transporters, also necessary for metabolism. Presumably, the other mitochondrial components especially are present in an optimal stoichiometric ratio with respect to the ATP synthase and may indeed occupy much more of the membrane area. For example, if the ATP synthase has an intrinsically higher enzymatic turnover frequency than the enzymes powering the generation of proton motive force, then it's amount can be comparatively small on a stoichiometric basis and the other membrane complexes may occupy a large fraction of the membrane surface.*

As noted below, in response to the second review, we have acknowledged the uncertainties in this area, but also note that protein packing issues will also apply to internal mitochondrial membranes (and perhaps even more so, owing to the need for proteins involved in the maintenance membrane folding). Thus, because there is not a dramatic increase in mitochondrial membrane area relative to that of the cell surface, the packing uncertainty does not seem to weaken our general conclusion that eukaryotes have not experienced a major increase in bioenergetics capacity relative to prokaryotes. Moreover, our goal throughout the paper has been to bring as many additional and independent lines of evidence to bear on this conclusion as possible – the smooth scaling of bioenergetics growth and maintenance requirements across the prokaryotic-eukaryotic divide, as well as the scaling of numbers of ATP synthase complexes and ribosomes, all support our general conclusion; and the substantial additional costs of building internal membranes in eukaryotic cells does as well.

*Reviewer #2:*

*[…] The authors revisit the hypothesis that the mitochondria were essential for the development of eukaryotic complexity for energetic reasons. The authors thoroughly analyze the ATP and other investments as performed by current eukaryotic cells and compare them to prokaryotes. They use empirical scaling laws to see if the observed changes are more than one would expect from simple scaling with cell volume. They find no strong evidence for a significant energetic benefit from mitochondria which leads them to cast doubt on high profile earlier reports.*

*I find the study scientifically sound and interesting. I have suggestions for improvement in terms of clarity and accuracy as given below.*

*Main text, third paragraph: "This implies that the mitochondrion-host cell consortium that became the primordial eukaryote did not precipitate a bioenergetics revolution."*

*In order to say it did not cause a bioenergetics revolution I need to have a definition of what is the definition such a revolution in as rigorous terms as possible. Either by the authors or by them repeating in detail a definition from previous authors.*

We sympathize with the reviewer’s request for more rigor here. The statements we have made are based on many made the Lane books, and also paraphrase the claims in the Lane and Martin paper. One could argue that these statements are a bit overstated and not based on any quantitative analysis, so is difficult to state them as formal hypotheses, but I think that we have come close to a representation in the first sentence in the section on “energy production in the mitochondrion”. Given the quotes we provide from the Lane and Martin paper below, it seems unlikely that any reader would find that we are overstating the claims being made. (The source of their repeated statements about a 200,000-fold expansion in genes and genome size eludes us, and makes no sense):

“The endosymbiosis that gave rise to mitochondria restructured the distribution of DNA in relation to bioenergetic membranes, permitting a remarkable 200,000-fold expansion in the number of genes expressed. This vast leap in genomic capacity was strictly dependent on mitochondrial power, and prerequisite to eukaryote complexity: the key innovation en route to multicellular life.”

“By enabling oxidative phosphorylation across a wide area of internal membranes, mitochondrial genes enabled a roughly 200,000-fold rise in genome size compared with bacteria. ……. Mitochondria increased the number of proteins that a cell can evolve, inherit and express by four to six orders of magnitude, but this requires mitochondrial DNA.”

“For four billion years bacteria have remained in a local minimum in the complexity fitness landscape, a deep canyon bounded on all sides by steep energetic constraints. The possession of mitochondria enabled eukaryotes to tunnel through this mountainous energetic barrier. Mitochondria allowed their host to evolve, explore and express 200,000-fold more genes with no energetic penalty.”

“Without mitochondria, prokaryotes – even giant polyploids – cannot pay the energetic price of complexity; …… The conversion from endosymbiont to mitochondrion provided a freely expandable surface area of internal bioenergetic membranes, serviced by thousands of tiny specialized genomes that permitted their host to evolve, explore and express massive numbers of new proteins in combinations and at levels energetically unattainable for its prokaryotic contemporaries. If evolution works like a tinkerer, evolution with mitochondria works like a corps of engineers.”

*Throughout the paper the scaling laws have no uncertainty ranges on their parameter values. This makes it hard to understand how predictive they are and should be corrected.*

These were given in our prior publication, and are now repeated here.

*Subsection “The energetic costs of building and maintaining a cell”, fourth paragraph: "that a shift of bioenergetics from the cell membrane in prokaryotes to the mitochondria of eukaryotes conferred no directly favorable energetic effects. In fact, the effect appears to be negative."*

*One could claim that because prokaryotic ATP production is associated with the cell membrane and it scales like the surface area an exponent of 1 with cell volume is not what one would expect (but rather 2/3) and the evidence supporting an approx ~1 exponent suggests there is some favorable energetic effect. I am not saying this is proof of such an effect but I think this point should be acknowledged/discussed.*

This is a good point that we had not made clear enough, so we now have added a sentence to this paragraph to make the SA:V expectation explicit.

*Subsection “Energy production and the mitochondrion”, last sentence: "and that the corresponding hypothetical packing density for eukaryotes would be 30% (if in the cell membrane)."*

*The authors do not seem to reflect more on this value they derive but it seems like a very high value to me. Given that packing of equally sized circles on a sphere cannot achieve more than I think about 60% usage of the sphere area this is not far from the maximal possible and this is before considering all the other protein machines needed in the membrane real estate or the requirements for lipids.*

These are good points, and we now make a statement just before “the biosynthetic cost…” section to this effect. We do not think that these uncertainties upset our general conclusions, as the more general and compelling evidence derives from the absolute surface areas of the cell vs. mitochondrial membranes, both of which will be subject to the same packing problems (and as noted, perhaps more in mitochondria).

“There are a number of uncertainties in these packing-density estimates, and more direct estimates are desirable. The optimum and maximum-possible packing densities for ATP synthase also remain unclear. Nonetheless, the fact remains that any packing problems that exist for the cell membrane are also relevant to mitochondrial membranes, which have additional protein components (such as those involved in internal-membrane folding).”

*Subsection “The biosynthetic cost of lipids”: "and Escherichia coli (… 0.98 μm^3^, respectively)"*

*The volume of an E. coli cell can easily change by a factor of 5 depending on growth rate so giving the volume as 0.98 μm^3^ without stating anything about growth conditions is odd. Better state as ~1 μm^3^ or the like.*

In general, we have reduced the numbers of digits used throughout, with no resultant changes in the conclusions.

*Discussion, fifth paragraph: "because the end state is slightly deleterious owing to the additional investment required to carry out individual tasks (Lynch et al. 2001)."*

*I found it hard to follow the logic here and I think other readers might have this problem. It is worth explaining in a bit more detail what is meant.*

This has been reworded in a way that is hopefully now clearer.

*Discussion, last paragraph: "It is plausible, that phagocytosis was a late-comer in this series of events, made possible only after the movement of membrane bioenergetics to the mitochondrion, which would have eliminated the disruptive effects of surface membrane ingestion on the ETC and ATP synthase."*

*I did not understand the connection here. Please clarify.*

We have tried to word this in a clearer way – the basic issue is that a cell would have a difficult time maintaining cell-membrane bioenergetics if the membrane and its resident ATP synthases was constantly being ingested.